# Personal Thermal Management by Single-Walled Carbon Nanotubes Functionalized Polyester Fabrics

**DOI:** 10.3390/ma14164616

**Published:** 2021-08-17

**Authors:** Liyuan Guan, Zhong Wang, Mingxing Wang, Yangjinghua Yu, Wenjian He, Ning Qi, Guohe Wang

**Affiliations:** 1College of Textile and Clothing Engineering, Soochow University, Suzhou 215006, China; 20174215007@stu.suda.edu.cn (L.G.); wangzhong1215@suda.edu.cn (Z.W.); 20174015009@stu.suda.edu.cn (Y.Y.); 2Violet Home Textile Science and Technology Co. Ltd., Nantong 226311, China; wangmingxingking@163.com; 3Suzhou Creation-Textile Co. Ltd., Suzhou 215000, China; wenjian@creation-textiles.com

**Keywords:** personal thermal management, single-walled carbon nanotubes, polyester fabrics, functionalize, electro-thermal performance

## Abstract

In this work, a personal thermal management (PTM) device based on single walled carbon nanotubes (SWCNTs) functionalized polyester fabrics had been studied. Polyester fabrics were functionalized with SWCNTs through coating method with poly (butyl acrylate) emulsion as the adhesive. The SEM images exhibited that SWCNTs formed high-efficiently conductive networks due to the large aspect ratio and uniform dispersion. A steady-state temperature of 40 °C was achieved at the input voltage of 2.5 V within 7 s, which exhibited excellent electro-thermal performance. Even under periodic heating-cooling conditions, heating system still displayed relatively stable temperature and relative resistance, which could have potential application for wearable clothes.

## 1. Introduction

Wearable devices, with various functions as health care, as personal thermal management (PTM), had become ubiquitous in daily life [1,2,3,4,5,6,7]. Certainly, light, comfortable and energy-efficient PTM devices are emerging and rapidly integrated into human lives. PTM devices could regulate body temperature to a thermally secure and comfort state by combining functional nano-materials with traditional textile, which could mitigate a vast amount of energy during the heating process, even could be helpful for the global warming and energy crisis.

Currently, different types of electro-thermal materials had been studied for using PTM device. Traditional electro-thermal materials as Cu, Fe and Cr based alloy show high thermomechanical properties, however some disadvantages such as low heating efficiency, heavy weight and uncomfortable feeling are far from expectation [8,9]. The application of indium tin oxide (ITO) with high conductivity is impeded due to the slow thermal response and frangibility [10,11,12,13]. Therefore, the considerable efforts are being devoted to the preparation of all kinds of excellent electro-thermal materials, such as metallic nanowire [14], graphene [15,16,17], conductive polymers [18,19,20] and carbon nanotubes (CNTs) [21,22].

CNTs had been discovered in 1991, with the properties of light, remarkable flexibility and excellent mechanical performance [23,24,25,26,27]. Better yet, the large aspect ratios of CNTs could shape 3-dimensional (3D) conductive networks, which greatly fit for the electrical energy converted into thermal energy with high-efficiently. From this viewpoint, CNTs could present a compliant material for the preparation of personal thermal management device. Single-walled carbon nanotube (SWCNTs), with outstanding electronic and mechanical performance, was most excellent and novel material among the CNTs family. Therefore, SWCNTs could be worked as part of ideal electro-thermal devices.

In this work, we proposed the strategy of developing the SWCNTs functionalized polyester fabrics as PTM device for heating system by coating method, while the heating system exhibited excellent electrical conductivity and electro-thermal properties with fast response and high stability, which could stable at 40 °C with 2.5 V, exhibit excellent potential on the electro-thermal devices.

## 2. Experimental Section

### 2.1. Materials and Pretreatment

SWCNTs viscous solution was supplied by Suzhou Institute of Nano-tech and Nano-bionics [28]. Polyester fabrics were obtained from Wujiang Fu Hua Weaving Co., Ltd., Suzhou, China. NaOH (chemical purity), butyl acrylate (chemical purity) and methacrylic acid (chemical purity) was purchased from Guoyao Company, Beijing, China. Styrene, primary alcobol Ethoxylate (AEO), sodium dodecyl sulfate, potassium persulfate and sodium bicarbonate (all chemical purity) was purchased from TCI Company, Tokyo, Japan.

The polyester fabrics were immersed in 7 g∙L^−1^ NaOH at 90 °C for 50 min to remove the impurity, then the polyester fabrics were washed under water until neutral and dried in an oven (Shanghai Jing Hong Laboratory Instrument Co.,Ltd, Shanghai, China) for one night.

The PBA emulsion was synthesized by the polymerization of butyl acrylate, methacrylic acid and styrene [29]. 2.04 g butyl acrylate, 0.87 g styrene and 0.07 g methacrylic acid was first added into a 500 mL flask, and stirred for 20 min. After that, 0.11 g AEO and 0.08 g sodium dodecyl sulfate was added into the flask and stirred to gain a pre-emulsion. Deionic water and sodium bicarbonate was added into the flask and heated to 75 °C. Then 10 mL of 2 g∙L^−1^ polymerization initiator potassium persulfate was dropped into the pre-emulsion with the spread of 2 drop∙s^−1^ and heated upon 80 °C for 30 min. The mixture was then heat to 85 °C for 1 h, and adjust Ph to neutral with the temperature of 60 °C. The PBA emulsion was collected after cool to the room temperature.

### 2.2. Fabrication of PTM Device

SWCNTs functionalized polyester fabrics were prepared by the coating method. The SWCNTs was homogeneous coated on a polyester fabric along the conductive threads by a mold with the length and width of 1 and 2.5 cm, and the coating concentration was 0.04 g·cm^−2^, respectively. Then, the sample was dried in a vacuum oven at 40 °C for one night. After that, the PBA emulsion was coated on the surface of the SWCNTs layer with the concentration of 16 μL·cm^−2^. Heating system equivalent model was exhibited in Figure 1 and Scheme 1. As an advanced stretchable structure, the island-bridge design has been widely explored for stretchable electronics, which could confine strains/de-formations to the stretchable interconnects [30]. Basically, the 9 pieces of SWCNTs coated on a polyester fabric (20 cm × 20 cm) were connected to a parallel circuit. Copper wire (0.2 mm) was used as the circuit for heating system through the polyester in a serpentine shape.

### 2.3. Characterization

The morphologies of carbon nanotubes functionalized fabrics were characterized by a field emission Scanning Electron Microcopy (SEM; S-4800, Hitachi, Tokyo, Japan) under high vacuum with the operation voltage of 3 kV. The Structure of SWCNTs was observed by high resolution transmission electron microscope (HRTEM; FEI TECNAI G20, FEI, Hillsboro, OR, USA). The SWCNTs dispersion was analyzed by testing Zeta potential (Nano-ZS90, Malvern, Malvin, UK) at 25 °C. Raman spectra was collected using a Micro-Raman Spectrometer (Labram Xplora, Horibajy, Paris, France) equipped with a 638 nm laser and a glass slide as substrate. Macroscopic morphology of coated surface of polyester was acquired by a digital microscope (VHX-100, Keyence, Osaka, Japan) with the magnification power of 150×. The electro-thermal experiments were examined under constant temperature and humidity to evaluate the heating performance of the heating system. The voltage was applied through a DC regulated power source (QJ6003S, QJE, Ningbo, China) from 0 and 3 V, and the crocodile clip test wire linked the wire of heating system to the power source. The resistance was measured using the digital multimeter (17b+, Fluke, Everett, WA, USA). Thermal imager (T620 2.1, FLIR, Wilsonville, OR, USA) was employed to monitor the surface temperature. The response time could be calculated by the relationship between temperature and time during the heating state. The washing stability was measured by the thermal performance with 0, 5 and 10 times washed fabric.

## 3. Results and Discussions 

### 3.1. Morphology and Dispersion of SWCNTs

The as-used CNTs with distinct layer number of 1 was SWCNTs, which could be observed from HRTEM image depicted in Figure 2a. The large aspect ratio of SWCNTs, which had given contribution to the formation of the chain-like and 3D conductive networks structure, could enrich the conduction pathway for electron transport and promote the long-distance transport of electrons (Figure 2b). However, SWCNTs were easily wrapped around by Van der Waals force owing to high specific surface. Then, the dispersion was measured by zeta potential. The zeta potential of SWCNTs aqueous dispersion was −34 mV. It indicated that the electron density accumulated on the surface was high and therefore produced large electrostatic force. It could greatly aid the dispersion of SWCNTs and facilitate the formation of outstanding electrically conductive pathway. The interfacial resistance was also reduced. It was contributed to acquire better electro-thermal performance.

SWCNTs, an electro-thermal material, was combined with polyester as a coating to form the heating system. From section SEM image of the coating fabric, the fracture surfaces could be observed directly, the top layer was the SWCNTs layer with the PBA emulsion (Figure 3a). Even though due to the concavo-convex of the fiber surface, the thickness of the layer was not extremely uniform. However, compared with the electro-thermal test, the SWCNTs emulsion layer was well-coated on the fabric surface. Due to the different observation of SWCNTs emulsion surface, the image of the heating system surface without emulsion was revealed by digital microscope in Figure 3b. The fabric could be observed to be uniformly coated with SWCNTs, which ensured the homogeneity of heating process.

### 3.2. Raman Spectra Analysis

Raman spectra had been observed to evaluate the SWCNTs coating on polyester fabrics (Figure 4). The characteristic bands at 148 and 194 cm^−1^, were named as radial breathing mode (RMB), which could be used as mark for single-walled. The peak at 1322 cm^−1^ was D-band that related to the defects and disorder of graphene sheets. [31] Tangential band (G-band) located at 1590 cm^−1^ correspond to C-C stretching. G’-band was Centered at 2630 cm^−1^, whose frequency shift was about twice as high as D-band, but the generation of G’-band was independent of the defects of graphite layer. Further calculated I_D_/I_G_ ratio was 0.12 on the basis of Figure 4, which suggested a high crystallinity of the SWCNTs.

### 3.3. Electro-Thermal Performance

Figure 5a exhibited the current-voltage characteristics for the heating system. The linear relationship between current and voltage was consistent. The current was increased with the voltage, and the correlation coefficient reach 0.99, which indicate the heating system could keep resistance stability within the adjusted voltage range.

The electro-thermal behaviors were measured under the constant temperature and humidity by inputting DC voltage between 0 and 3 V (Figure 5b). The conversion mechanism could be explained by the fact that the migration of charge carriers in the system might become accelerated due to the influence of external electric potential. These accelerated electrons might collide in-elastically with phonons, impurities or defects presented in the SWCNTs walls, which lead to the heat release [32,33].

The heating temperature was controlled at 40 °C due to the comfort for the human body. The quadratic correlation of the maximum temperature (T_max_) and voltage were presented in Figure 5b. The T_max_ increased with the increasing voltage, which certificated that the heating system had excellent electro-thermal property owing to the coating of SWCNTs. Consequently, the eletro-thermal behavior of heating system prepared could be controlled effectively by adjusting the input voltage. A steady-state temperature of 38 °C was gained under 2.4 V, and it could promote to 43 °C by increasing the voltage up to 2.6 V. It could be inferred that the required temperature of 40 °C could be obtained around the voltage of 2.5 V. Better heating performance could be obtained at lower voltage, which demonstrates efficient and safety of the electro-thermal conversion of heating system.

The time-dependent temperature curves could be divided into three main regions in Figure 6, the heating region, the steady region and the cooling region. The temperature increased rapidly and reached the steady-state temperature of 40 °C with 2.5 V from the room temperature. During this stage, only a small portion Joule heat was lost into the surroundings, and the most was used to heat the system. In the steady temperature region, heat converted from electric energy was equilibrated with the heat that lost into surroundings by convection and radiation, which based on the energy conversation law. At last, heating system cooled rapidly to ambient temperature as the power source was turned off at 200 s.

Moreover, the response time, which could be defined as the time required to reach 90% of the steady-state temperature, is one of the main factors to evaluate the electro-thermal performance of heating system [31]. According to the calculation, the response time was less than 7 s. The results illustrated that heating system could display a rapid electro-thermal response to the applied voltage.

The operational stability was an important parameter for the electric heating element which determines the service life and market prospect. The cycling heating-cooling tests were conducted to represent the long-term stable state. The tests were performed 10 times under periodic input voltage of 2.5 V, with an on/off-ratio of 300 s (Figure 7a). In General, the whole cyclic process maintained faster heating, cooling responses and steady-state temperature fluctuated around 40 °C, which supported the view that the heating system prepared has excellent repeatable electrical heating performance.

Additionally, the electro-thermal stability was quantitatively measured by the relative resistance R/R_0_ (R is the initial resistance, R_0_ is the resistance in different cycles) which could reveal the electro-thermal reproducibility. Figure 7b exhibited that the T_max_ and R/R_0_ were slightly changed with the increasing cycle number. Initially, the T_max_ rose and the R/R_0_ decreased due to the warmup, and then remained stable in general during the heating-cooling cycles. consequently, it could be inferred that the heating system had high thermal stability and advanced application prospect in PTM devices.

### 3.4. Washing Stability

The washing stability was measured by the thermal performance with the fabric washed for 0, 5 and 10 times. The fabric was washed in a oscillating machine with a soap concentration of 0.04 g·L^−1^ at a temperature of 40 °C for 30 min. The results were exhibited in Figure 8. All of the fabric samples exhibit a rapid temperature response at the first 20 s while the stable temperature was about 40 °C. The fabric samples exhibit nearly the same curve of the thermal performance, which proves that this system was quite stable in the washing process.

## 4. Conclusions

A personal thermal management device based on single-walled carbon nanotubes functionalized polyester fabrics was successfully manufactured and the electro-thermal performance was systematically investigated. The SEM image and zeta potential indicated that SWCNTs were uniform dispersed, which reduce interfacial resistance and produced better conductive networks. The excellent electro-thermal performance was also observed. Heating system revealed an evidently rapid temperature response to the voltage, and in the continuous heating-cooling cycles test, T_max_ and R/R_0_ remained stable in general. In brief, SWCNTs functionalized polyester fabrics, as the heating system, exhibited rapid electro-thermal response and operational stability. Moreover, PTM could reduce individual demand for power and lessen dependence on energy. From the aforementioned viewpoint, SWCNTs functionalized polyester fabrics could be used for the intelligent heating clothing with a controllable and safe temperature.

## Data Availability

The data presented in this study are available on request from the corresponding author.

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
