# Peer review of "Personal Thermal Management by Single-Walled Carbon Nanotubes Functionalized Polyester Fabrics"

_materials, 2021, doi:10.3390/ma14164616_

Round 1
Reviewer 1 Report
Dear Authors/Editor,
Thank You very much for the trust and opportunity to revise scientific paper entitled: “Personal thermal management by single-walled carbon nano[1]3 tubes functionalized polyester fabrics” written by Liyuan Guan, Zhong Wang, Mingxing Wang, Yangjinghua Yu, Wenjian He, Ning Qi and Guohe Wang.
In my opinion the article has a good scientific level and the content is very interesting. The topic is very modern and valuable for the special application in industry. It is written in a understandable style and the results are quite well documented. The literature is actual and well selected. So I recommend to publish this paper after small corrections.
Here you have some comments and suggestions for authors:
Page 2:
- Line 59: ….alcobol Ethoxylate (AEO) - I think it is alcohol?
- Line 64-65: “The PBA emulsion was synthesized by the polymerization of butyl acrylate, 65 methacrylic acid and styrene. [29] “ - Please add more details how the synthesis was carried out, molar ratio, conditions, initiators etc.
- Line 77-78: Copper wire (0.2 mm) was used as the circuit for heating system, through the polyester 78 in a serpentine shape. – There is only a schematic view but I think that real image of the prepared material should be inserted here.
Page 5, line 146: “3.3 electro-thermal performance” The chapter title should start with capital letter.
Page 9, References –Some page ranges are missed for: 4, 6, 21, 29, 30 ,32.
Best regards.
Author Response
Dear reviewer, thank you for your kindly reminds and questions. All of the questions had been answered as follow:
Q1: We had check the name of it. Its name was right with the CAS no. of 68131-39-5.
Q2: “2.04 g butyl acrylate, 0.87 g styrene and 0.07 g methacrylic acid was first added into a 500 ml flask, and stirred for 20 min. After that, 0.11g AEO and 0.08 g sodium dodecyl sulfate was added into the flask and stirred to gain a pre-emulsion. Deionic water and sodium bicarbonate was added into the flask and heated to 75 ℃. Then 10 mL of 2 g∙L-1 polymerization initiator potassium persulfate was dropped into the pre-emulsion with the spead of 2 drop∙s-1 and heated upon 80 ℃ for 30 min. The mixture was then heat to 85 ℃ for 1 h, and adjust pHto neutral with the temperature of 60 ℃. The PBA emulsion was collected after cool to the room temperature.” had been added to explain the synthesize of PBA emulsion.
Q3: an image of the real sample was added as figure 1(b) in the manuscript.
Q4: We had correct the errors of this.
Q5: All of the reference had been checked and corrected with only page number and other errors.
Reviewer 2 Report
The manuscript presents very interesting and well described results. Authors used many different methods for investigations.
I believe that the manuscript can be published as it is.
Author Response
dear reviewer, thank you for your kindly review.
Reviewer 3 Report
The manuscript presents some significant and interesting results on CNT-polyester fabrics for personal thermal management.
The manuscript can be accepted after considering the following points:
- English language should be improved throughout the manuscript.
- Section 2.1: Materials and methods: The synthesis of PBA emulsion should be provided. A reference was given but some brief details are needed.
- The biggest issue with this paper is the originality/novelty of the work. There has been several reports (https://www.sciencedirect.com/science/article/pii/S0264127515307917) and many more on CNT and polyester based conductive fabrics. So, the authors should clearly indicate the novelty of their work.
Author Response
Dear reviewer, thank you for your kindly reminds and questions. All of the questions had been answered as follow:
Q1: English language correction had been down.
Q2: “2.04 g butyl acrylate, 0.87 g styrene and 0.07 g methacrylic acid was first added into a 500 ml flask, and stirred for 20 min. After that, 0.11g AEO and 0.08 g sodium dodecyl sulfate was added into the flask and stirred to gain a pre-emulsion. Deionic water and sodium bicarbonate was added into the flask and heated to 75 ℃. Then 10 mL of 2 g∙L-1 polymerization initiator potassium persulfate was dropped into the pre-emulsion with the spead of 2 drop∙s-1 and heated upon 80 ℃ for 30 min. The mixture was then heat to 85 ℃ for 1 h, and adjust pHto neutral with the temperature of 60 ℃. The PBA emulsion was collected after cool to the room temperature.” had been added to explain the synthesize of PBA emulsion.
Q3: Indeed, many researchers are studying on carbon nanofibers which had the benefit as light, remarkable flexibility and excellent mechanical performance. First of all, most researches were focused on the multi-walled CNTs (MWCNTs), only a few research had been down on the SWCNTs. SWCNTs might have the most effective and excellent performance among the CNTs family. While for our research, we are focus on the application as a personal thermal management, which should be suitable for human body. The controllability and effectiveness of the SWCNTs is suitable for it. So we choose the SWCNTs as the research object.
